# Inhibition of CBP synergizes with the RNA-dependent mechanisms of Azacitidine by limiting protein synthesis

Jeannine Diesch[1,2], Marguerite-Marie Le Pannérer [1], René Winkler [1], Raquel Casquero[1], Matthias Muhar[3], Mark van der Garde[4,5], Michael Maher[1], Carolina Martínez Herráez [6,7], Joan J. Bech-Serra[1], Michaela Fellner[3], Philipp Rathert [3,11], Nigel Brooks[8], Lurdes Zamora[9], Antonio Gentilella [6,7], Carolina de la Torre [1], Johannes Zuber [3,10], Katharina S. Götze[4] & Marcus Buschbeck [1,2✉]

The nucleotide analogue azacitidine (AZA) is currently the best treatment option for patients with high-risk myelodysplastic syndromes (MDS). However, only half of treated patients respond and of these almost all eventually relapse. New treatment options are urgently needed to improve the clinical management of these patients. Here, we perform a loss-of-function shRNA screen and identify the histone acetyl transferase and transcriptional co-activator, CREB binding protein (CBP), as a major regulator of AZA sensitivity. Compounds inhibiting the activity of CBP and the closely related p300 synergistically reduce viability of MDS-derived AML cell lines when combined with AZA. Importantly, this effect is specific for the RNA-dependent functions of AZA and not observed with the related compound decitabine that is only incorporated into DNA. The identification of immediate target genes leads us to the unexpected finding that the effect of CBP/p300 inhibition is mediated by globally down regulating protein synthesis.

[1] Cancer and Leukaemia Epigenetics and Biology Program, Josep Carreras Leukaemia Research Institute (IJC), Campus ICO-GTP-UAB, Badalona, Spain. [2] Program for Predictive and Personalized Medicine of Cancer, Germans Trias i Pujol Research Institute (PMPPC-IGTP), Badalona, Spain. [3] Research Institute of Molecular Pathology (IMP), Vienna BioCenter (VBC), Vienna, Austria. [4] Department of Medicine III, Klinikum rechts der Isar, Technische Universität München, Munich, Germany. [5] German Cancer Consortium (DKTK), Partner Site Munich, Heidelberg, Germany. [6] Laboratory of Cancer Metabolism, ONCOBELL Program, Institut d'Investigació Biomèdica de Bellvitge (IDIBELL), L'Hospitalet de Llobregat, Spain. [7] Faculty of Pharmacy, Department of Biochemistry and Physiology, Universitat de Barcelona, Barcelona, Spain. [8] CellCentric, Ltd, Chesterford Research Park, Little Chesterford, Cambridge CB10 1XL, UK. [9] Hematology Laboratory Service, ICO Badalona-Hospital Germans Trias I Pujol, Josep Carreras Leukemia Research Institute (IJC), Badalona, Spain. [10] Medical University of Vienna, Vienna BioCenter (VBC), Vienna, Austria. [11]Present address: Biochemistry Department, University Stuttgart, Stuttgart, Germany. ✉email: mbuschbeck@carrerasresearch.org

Myelodysplastic syndromes (MDSs) are a heterogeneous group of clonal hematopoietic stem cell diseases characterized by dysplasia and ineffective haematopoiesis affecting one or more myeloid cell lineages[1]. MDSs are part of a spectrum of myeloid diseases that evolve from asymptomatic clonal haematopoiesis with indeterminate potential and transform into secondary acute myeloid leukemia (sAML) in 30–40% of cases[2]. Environmental or therapy-related factors accelerate the development of MDS[3,4]. MDS are among the most frequent haematological disorders in the elderly[5] and sAML has particularly poor prognosis[6]. The genetic alterations in MDSs are well described and include mutations in genes involved in epigenetic regulation and RNA processing[7,8].

Currently, the azanucleosides azacitidine (AZA) and decitabine have multiple clinical applications and are, for instance, the alternative standard treatment options for high-risk MDS patients that are not eligible for an allogeneic transplant[9]. However, the response is only partial: 40–50% of treated patients show haematological improvements and a complete response is limited to as few as 10–15%[9,10]. Furthermore, AZA is also used for the supportive treatment of a subset of AML cases[11]. Azanucleosides are frequently referred to as hypomethylating agents and epigenetic drugs, as they interfere with DNA methylation[12]. In additon to reducing global DNA methylation, azanucleosides are general nucleotide analogues that affect multiple cellular functions. After cellular uptake, azanucleosides are metabolically converted into their active forms and incorporated into newly synthesized DNA and RNA[13]. AZA and decitabine differ in that decitabine is exclusively incorporated into DNA, whereas 80–90% of AZA becomes part of RNA[14]. Incorporation into DNA leads to the formation of covalent adducts between DNA and DNA methyltransferases (DNMTs), particularly DNMT1[15], for which there are two consequences. First, these adducts elicit the DNA damage response in an ataxia-telangiectasia mutated (ATM)/ataxia telangiectasia and Rad3-related protein (ATR)-dependent manner and, if not properly repaired, cause cytotoxicity[16]. Second, the inhibition of the cross-linked DNMTs leads to global DNA hypomethylation that induces the expression of a subset of genes promoting cell differentiation[17]. In addition, the hypomethylation-induced transcription of endogenous retroviruses mimics a viral infection and activates the innate immune response[18].

The incorporation of AZA into RNA has additional effects. Analogous to the inhibition of DNMTs, AZA inhibits a subset of RNA methyltransferases that methylate position 5 of cytosine in transfer RNAs (tRNAs) and other RNAs[19,20]. AZA sensitivity is best illustrated for DNMT2[21]. As a consequence, AZA but not decitabine inhibits protein synthesis[22] and this is at least in part mediated by tRNAs[23]. AZA also impacts on messenger RNA (mRNA) stability. In this context, AZA treatment leads to the downregulation of ribonucleotide reductase by destabilizing the mRNA encoding an essential component of the enzymatic complex[24]. Thereby, AZA reduces the conversion of ribonucleotides to deoxyribonucleotides required for DNA synthesis, which includes its own conversion to decitabine[24]. Furthermore, AZA disrupts the RNA-dependent interaction of the complex containing hnRNPK, DNMT2 and NSUN3 with Polymerase 2 on actively transcribed genes[25]. Taken together, these additional RNA-dependent functions of AZA provide part of an explanation for why AZA and decitabine cause largely non-overlapping changes in mRNA levels[22,26].

The CREB-binding protein (CBP) and its paralogue p300 are highly conserved transcriptional co-activators that are involved in many physiological processes, such as proliferation, differentiation and apoptosis[27,28]. CBP and p300 have a high degree of homology and their functions are largely redundant[29]. Their regulating properties can be attributed to three main functions. First, they bridge sequence-specific DNA-binding proteins to the basal transcription machinery. Second, they act as transcriptional co-activators of a large number of transcription factors. Finally, they are enzymes with lysine acetyltransferase activity that acetylate histone and non-histone proteins. In haematopoiesis, CBP and p300 contribute to cell differentiation and stem cell maintenance through their interaction with key transcription factors such as GATA1[30], MYB[31] and AML-1[32]. In AML and chronic myelomonocytic leukemia (MLL) translocations of MLL to CBP are common, leading to gain-of-function of MLL[33]. Similarly, MOZ-CBP and MOZ-p300 translocations can drive AML leukemogenesis[34]. In addition, many oncogenes interact with CBP and p300, making it important factors in malignant transformation[27]. The deregulation of CBP and p300 in blood cancers has prompted the development of inhibitors[35] and one of these, CCS1477, has entered clinical development in an ongoing first in human trial in haematological cancers (NCT04068597).

Here we report the results of a genetic loss-of-function screen of chromatin regulators that leads to the identification of CBP as a combinatorial drug target for AZA-based treatments in vitro. We dissect the synergistic interaction between AZA and pharmacologic CBP/p300 inhibitors, and are able to link it to the RNA-dependent functions of AZA. Moreover, we demonstrate a previously unrecognized but prominent interference with protein synthesis by CBP/p300 inhibitors.

## Results

**An improved genetic loss-of-function screen identifies chromatin regulators that affect the cell-intrinsic response to AZA.** The need for improvement of the response rate and duration of AZA-based therapy has motivated us to search for potential combinatorial drug targets. As AZA is a chromatin-modifying agent[11], we set out to determine the effect of inhibiting chromatin regulators on treatment response in vitro. We have opted for an RNA interference-mediated knockdown approach that is more similar than a knockout approach to the level of inhibition that can be expected to be reached with drugs in the clinic. As a cellular model, we have chosen the MDS-derived sAML cell line SKK-1 that has been well characterized in our laboratory[36]. SKK-1 cells have been isolated from a patient with MDS-derived sAML and harbour mutations in splicing factor U2AF1 and the chromatin regulator BCOR[37]. To test chromatin regulators in a most comprehensive manner, we have generated a lentiviral library targeting 912 human chromatin regulatory genes with 8 independent short hairpin RNAs (shRNAs) per gene (Supplementary Data 1 and Fig. 1a). shRNAs were embedded in an optimized microRNA backbone, allowing efficient knockdown of target genes after single-copy integration[38,39]. To ensure efficient shRNA expression and prevent transgene repression, we further optimized the previously described lentiviral pRRL-SFFV-GFP-miRE-PGK-Puro (SGEP) backbone[38] by inserting a ubiquitous chromatin opening element (UCOE, Fig. 1b). The UCOE originates from the CBX3 promoter and has anti-silencing properties[40]. We have tested the functionality of the modified vector, termed cSGEP, by confirming the cytotoxic effects of shRNAs targeting essential genes (Supplementary Fig. S1a). Specifically, we have targeted the transcriptional coactivator BRD4, the hematopoietic transcription factor MYB and the core sub-unit of RNA polymerase 2 RPA3, which are known to be essential in AML cell lines.

For the loss-of-function screening, SKK-1 cells were infected with the cSGEP shRNA library at low viral titre to favour single-copy integration (Fig. 1c), puromycin selected and half of the cells were treated for 2 weeks with partial lethal doses of AZA that we

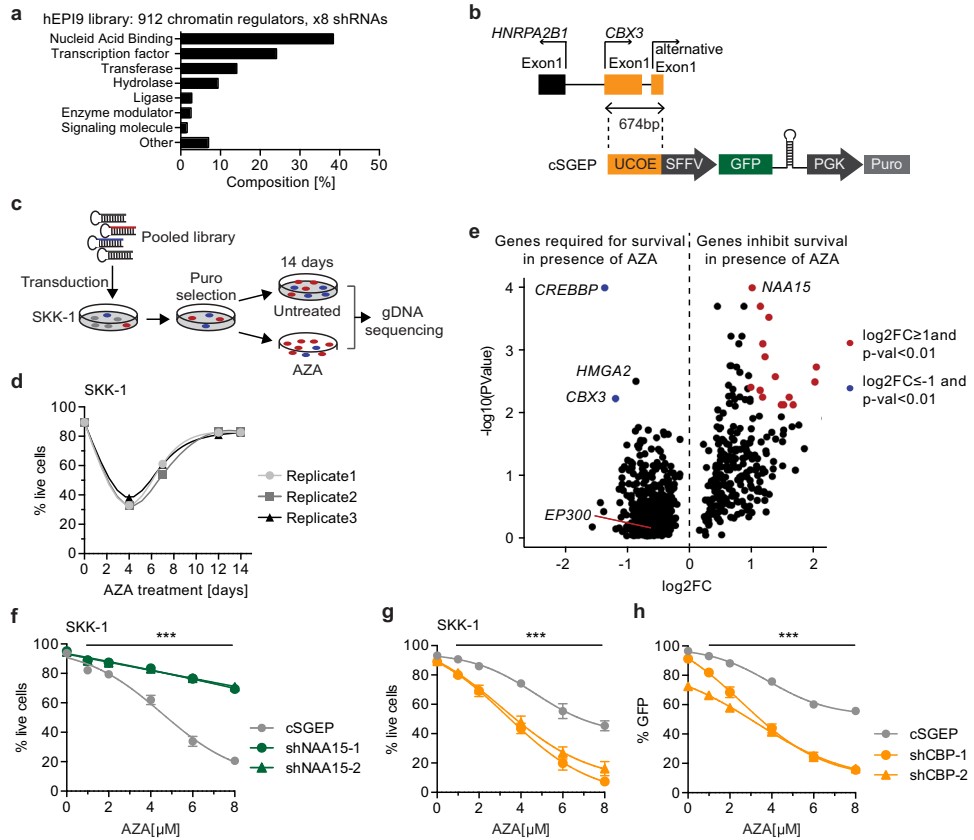

**Fig. 1 Chromatin regulators affecting AZA sensitivity are identified by an optimized loss-of-function shRNA screen. a** Composition of the hEPI9 library. **b** Illustration of the human *HNRPA2B1-CBX3* locus including the ubiquitous chromatin opening element (UCOE) and schematic of the cSGEP backbone vector containing the UCOE, spleen focus-forming virus (SFFV) promoter, green fluorescent protein (GFP) marker gene, shRNA integration site, phosphoglycerate kinase (PGK) promoter and puromycin (Puro) resistance gene. **c** Schematic workflow of the shRNA screen-targeting genes encoding chromatin regulators. **d** Percentage of live SKK-1 cells determined at different time points throughout the shRNA screen after initiation of AZA treatment (0.25 μM every 2 days for 2 weeks) in three biological replicates. **e** Volcano plot of genes corresponding to enriched ("Genes inhibit survival in presence of AZA") or depleted ("Genes required for survival in presence of AZA") shRNAs. Selected genes are named. FC, fold change. CREBBP, $P = 0.0001$; NAA15, $P = 0.0001$; HMGA2, $P = 0.0032$; CBX3, $P = 0.0061$; EP300, $P = 0.7405$. **f** Percentage of live SKK-1 cells stably expressing cSGEP, shNAA15-1 or shNAA15-2 determined after 4 days of treatment with indicated concentrations of AZA. **g**, **h** SKK-1 cells stably expressing cSGEP, shCBP-1 or shCBP-2 were treated for 4 days with indicated concentrations of AZA and the percentage of live cells (**g**) or the percentage of GFP-positive cells was assessed (**h**). **f–h** Data represent the mean ± SEM of four independent experiments. Statistical analysis was performed using two-way ANOVA. ***p-value < 0.0001. Source data are provided as a Source Data file and in Supplementary Data 1 and 3.

previously determined in dilution series (Supplementary Fig. S1b). Treatment of cells led to a rapid reduction in the percentage of live cells after 4 days followed by subsequent recovery, suggesting the expansion of cell clones with a growth and survival advantage (Fig. 1d). To identify genes affecting the response to AZA, we have compared the abundance of all shRNA clones in control and treated cells by massive parallel sequencing of the transgenes in genomic DNA purified from cell populations at the end of the treatment. To identify relevant genes, we only considered those genes for which at least five of the eight shRNAs showed the same trend and calculated their average fold change (Supplementary Data 3). Using this approach, we were able to identify 451 genes, whose shRNAs were significantly depleted, and 252 genes, whose shRNAs were enriched (Fig. 1e). A decrease of shRNAs in the treated cells indicated that the knockdown of the corresponding genes was not tolerated, and that the gene products were required for survival in the presence of AZA treatment. Among the top hits were *CREBBP* encoding for the transcriptional coactivator CBP. Conversely, genes associated with increased sensitivity to AZA were presented by enrichment of their shRNAs, as their knockdown reduced the sensitivity and thus increased the survival of the cells in the presence of AZA. This included the

gene encoding for the *N*-α-acetyltransferases, NAA15, which is a component of the NatA complex binding to ribosomes and co-translationally acetylating proteins[41].

Depleted hairpins were of particular interest to us, as they indicate potential combinatorial drug targets. Thus, we selected the top depleted hit encoding for CBP for further analysis and included the top enriched gene *NAA15* as control. To validate the results of the shRNA screen, we generated polyclonal SKK-1 cell lines with stable single knockdowns of CBP or NAA15 selecting two independent shRNAs per gene (Supplementary Fig. S1c, d). We assessed viability of these cells after 4 days of treatment with various concentrations of AZA ranging around the previously calculated half maximal inhibitory concentration (IC50) (Supplementary Fig. S2a). In line with the results of the shRNA screen, we observed a pronounced increase in viability in the NAA15-deficient cells reflecting decreased sensitivity to the drug (Fig. 1f). CBP-deficient cell lines showed the opposite behaviour with decreased cell viability in the presence of AZA (Fig. 1g). This increased sensitivity of cells with the CBP knockdown was also visible when monitoring the percentage of infected and non-infected cells among viable cells grown together. In such a competitive growth condition, CBP knockdown cells depleted

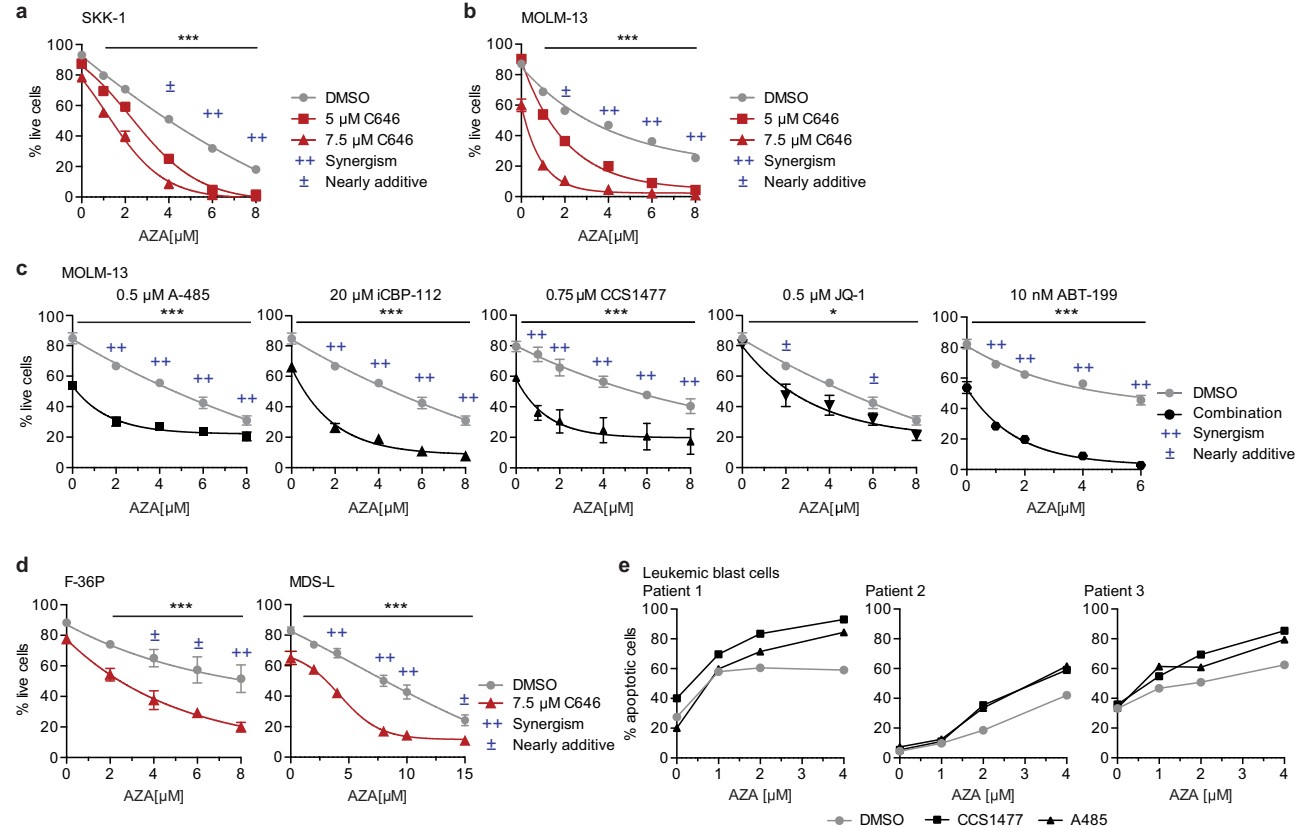

**Fig. 2 Inhibition of CBP/p300 increases sensitivity towards AZA in AML cells. a**, **b** Percentage of live SKK-1 (**a**) or MOLM-13 cells (**b**) determined by DAPI/MitoTracker staining after 4 days of treatment with indicated concentrations of AZA in combination with C646 or 0.075% DMSO as vehicle control. **c** Percentage of live MOLM-13 cells after 4 days of treatment with different concentrations of AZA in combination with indicated concentrations of the CBP/p300 inhibitors A-485, iCBP-112 and CCS1477, the BET bromodomain inhibitor JQ-1, the BCL-2 inhibitor ABT-199 or 0.03% DMSO as vehicle control. **d** Percentage of live F-36P or MDS-L cells after 4 days of treatment with indicated concentrations of AZA in combination with C646 or DMSO (0.075%) as vehicle control. **e** Percentage of apoptotic leukemic blast cells from three AML patients determined by Annexin V staining after 3 days of treatment with different concentrations of AZA and 2 days with 500 nM CCS1477 or 8 µM A-485, or 0.08% DMSO as vehicle control. **a**–**d** Combination index (CI) was calculated according to the Chou–Talalay method[43]. ++ synergism (CI: 0.3–0.85), ± additive effect (CI: 0.9–1.1). Data represent the mean ± SEM of four independent experiments. Statistical analysis was performed using two-way ANOVA. *$p$-value < 0.05; ***$p$-value < 0.0001. Source data are provided as a Source Data file.

more rapidly in the presence of AZA than control cells (Fig. 1h). Accordingly, we observed an increase in apoptosis in CBP knockdown cells treated with AZA (Supplementary Fig. 1e).

Taken together, employing a genetic loss-of-function screen of chromatin regulators allowed the identification of several genes affecting the sensitivity to AZA in SKK-1 cells. Specifically, we identified the histone acetylase CBP as a potential drug target for combinatorial treatments.

**Inhibition of CBP/p300 increases sensitivity towards AZA**. To further validate the impact of CBP on AZA sensitivity, we took advantage of the availability of small compound inhibitors of CBP/p300. The molecule C646 was the first inhibitor of CBP and p300 to be developed, and functions by competing with the co-substrate acetyl-CoA for binding to the enzymatic KAT domain[42]. We first determined the IC50 for C646 in SKK-1 cells (Supplementary Fig. S2a). To assess the combinatorial effect of C646 with AZA, cells were treated with a range of AZA concentrations alone or in combination with C646 at a concentration close to its IC50 or slightly below. Similar to the effect of CBP knockdown, we could observe a significant reduction in cell viability in response to the combination of C646 with AZA in comparison to AZA treatment alone (Fig. 2a). Calculating the combination index (CI) using the Chou–Talalay method[43]

indicated a synergistic interaction between both compounds (Fig. 2a). This was also reflected in a substantial increase in the number of apoptotic cells when treating with a combination of both drugs (Supplementary Fig. S3a). C646 treatment also further increased the response of CBP-depleted cells, suggesting that the inhibition of p300 or residual CBP contributed to the observed effect (Supplementary Fig. S3b).

We next assessed the effects of CBP/p300 inhibition on AZA response in an additional cell line. MOLM-13 is another MDS-derived sAML cell line but carries different driver mutations than SKK-1[36]. These include some genetic alterations that are more prevalent in other AML subtypes such as FLT3 internal tandem repeats, an MLL-AF9 fusion and mutations in *CBL*, *KMT2A* and *NF1* genes. As MOLM-13 cells are more commonly used in in vitro experiments and their characteristics are more comparable to the ones seen in MDS and sAML patients, we decided to further focus on this cell line. First, we determined the IC50 for AZA and C646 in cell viability assays (Supplementary Fig. S2b). Similar to the results in SKK-1 cells, we observed a significant and synergistic reduction in cell viability when combining C646 and AZA (Fig. 2b). Again, this was also reflected by increased apoptosis in response to the combinatorial treatment compared to the individual treatments (Supplementary Fig. S3c). Next, we tested additional CBP/p300 inhibitors in combination with AZA in MOLM-13 cells. For this, we

chose the potent and highly specific KAT domain inhibitor A-485[44] and the CBP/p300-specific bromodomain inhibitors iCBP-112 and CCS1477[45,46]. Importantly, CCS1477 is the first inhibitor of CBP/p300 to be tested first in human clinical trials in AML, multiple myeloma and non-Hodgkin lymphoma (NCT04068597), and in castration-resistant prostate cancer (NCT03568656)[47]. We further included the more general Bromodomain and extraterminal domain (BET) inhibitor JQ-1, which inhibits among others the bromodomains of downstream effectors BRD2 and BRD4[48]. As a benchmark and reference, we used the BCL-2 inhibitor ABT-199 (venetoclax)[49], which is US Food and Drug Administration (FDA) approved, in combination with AZA or decitabine for the treatment of AML in adults of 75 years or older (https://www.fda.gov/drugs/resources-information-approved-drugs/fda-grants-regular-approval-venetoclax-combination-untreated-acute-myeloid-leukemia). We determined the IC50 values for these compounds in MOLM-13 cells (Supplementary Fig. S2c). Combination of AZA with concentrations of A-485, iCBP-112 and CCS1477 close to their IC50 demonstrated a significant decrease in cell viability and synergy with AZA (Fig. 2c). Notably, the combination of AZA with CBP inhibitors was more effective than the combination with the BET inhibitor JQ-1 and in a similar range as the combination with ABT-199 (Fig. 2c). We further confirmed the increased response to AZA in the two additional sAML cell lines, F36-P and MDS-L, co-treated with C646 (Fig. 2d). Next, we used primary AML blasts isolated from three independent patients (patient characteristics in Supplementary Data 2) and treated them with A-485 and the clinically more relevant inhibitor CCS1477. In all three cases, the combination was more effective than the AZA mono-treatment (Fig. 2e).

In summary, these results indicate that inhibitors of CBP and p300 increase the response of AML cells to AZA. Remarkably, the combination reaches a similar synergy in cell lines as the FDA-approved combination of AZA and ABT-199.

**SLAM-seq identified direct target genes of CBP/p300 relevant for protein synthesis**. Although AZA has been widely used and studied during the last decades, CBP/p300 inhibitors have only recently gained clinical interest and the biological consequences of inhibiting CBP/p300 are not fully known. To further explore the effects of CBP/p300 inhibition and to understand the mechanism causing the synergistic effect of AZA and C646, we set out to identify the direct target genes of CBP/p300. For this, we quantified newly synthesized mRNA after short-term inhibition of CBP/p300 and metabolic labelling of mRNA with a method termed thiol(SH)–linked alkylation for the metabolic sequencing of RNA (SLAM-seq)[50]. Specifically, we analysed MOLM-13 cells treated or not treated with C646 for 2 h coinciding with metabolic labelling during the last hour (Fig. 3a). Calling differentially expressed genes from reads of newly synthesized transcripts defined by T > C conversions, we were able to detect a prominent and broad repression of nascent transcripts in treated cells (Fig. 3b, right panel) (Supplementary Data 4). This was not detectable when analysing all reads and plotting global mRNA levels (Fig. 3b, left panel) (Supplementary Data 4). After applying stringent cutoffs (false discovery rate (FDR) ≤ 0.01, log2FC ≤ −1/log2FC ≥ 1) to the newly synthesized transcripts, we identified 220 upregulated genes and 2253 downregulated genes in C646-treated cells compared to the dimethyl sulfoxide (DMSO) control. The strong over-representation of downregulated genes was in line with the role of CBP/p300 as transcriptional coactivator. By performing functional Gene Ontology analysis on the downregulated genes, we found an enrichment of genes that are important for RNA biogenesis and protein synthesis (Fig. 3c). Gene ontologies related with protein folding, heat shock proteins and chaperone binding and protein stabilization were enriched among the tenfold smaller set of upregulated

genes (Fig. 3d) (Supplementary Data 5). The inhibition of genes involved in protein translation and the upregulation of chaperone genes could be confirmed by real-time quantitative PCR (RT-qPCR) after 4 h of C646 treatment (Fig. 3e, f). Importantly, inhibition of CBP/p300 using another more specific inhibitor, A-485, confirmed the downregulation of genes involved in protein translation (Fig. 3g). The inhibition of a subset of genes in response to C646 could be further validated in SKK-1 cells (Fig. 3h).

Taken together, our results suggest that in MDS-derived sAML cells, CBP/p300 promotes the expression of genes involved in protein synthesis and cellular homoeostasis.

**Inhibition of CBP/p300 limits protein synthesis**. Through its incorporation in RNA, AZA has previously been shown to inhibit protein synthesis[13]. This led us to wonder whether the synergistic effect of CBP/p300 inhibition with AZA in cell viability could be due to potentiating the RNA-dependent function of AZA and by limiting protein synthesis to a level that is not tolerable for leukemia cells. In order to test this intriguing hypothesis, we analysed the combination of the CBP/p300 inhibitor C646 with the AZA analogue decitabine. In contrast to AZA, its analogue decitabine is not incorporated into RNA and hence does not affect protein synthesis[22]. Indeed, we found that the combination of C646 and decitabine on cell viability of MOLM-13 and SKK-1 cells was neither additive nor synergistic and, in the case of SKK-1 cells, even strongly antagonistic (Fig. 4a). To make sure that SKK-1 and MOLM-13 cells lines are responsive to decitabine treatment, we tested its inhibitory effect on LINE-1 DNA methylation (Supplementary Fig. S4).

Next, we wished to better understand to which extent CBP/p300 inhibition affects protein synthesis. Similar to SLAM-seq for detecting mRNA, we used metabolic labelling to identify and quantify newly synthesized proteins by mass spectrometry (MS, Fig. 4b). Specifically, we have chosen a short treatment with C646 of 4 h to avoid secondary effects such as those from apoptotic signalling. As shown in Fig. 4c, we observed a prominent reduction of the total number of detected nascent proteins from 2807 to 638 (average of replicates) by CBP/p300 inhibition (Supplementary Data 6). Only 160 proteins were detected in the background control without labelling and were excluded from further analysis, with the exception of a few proteins that were detected at 10-fold or higher level in untreated samples. Furthermore, we decided to focus our analysis on the 2423 proteins that were detected in all DMSO control samples. The number of peptide spectrum matches (PSMs) per protein was reduced after CBP/p300 inhibition, indicating a reduction in abundance of detected proteins (Fig. 4d). Using a reduction in PSMs of 75% or more as cutoff indicated that 2125 proteins, corresponding to 68% of all proteins detected in DMSO samples, were affected by CBP/p300 inhibition (Fig. 4e). When comparing the SLAM-seq (Fig. 3b) and proteomics results, we found partial overlap of 29% and 28% of the proteomics and SLAM-seq hits, respectively, between newly synthesized mRNA transcripts and corresponding newly synthesized proteins following CBP/p300 treatment (Fig. 4f). This suggests that the impact of CBP/p300 inhibition on protein synthesis cannot be solely attributed to a reduction of mRNAs but rather to a global downregulation of the protein translation machinery. Closer inspection of the list of affected proteins indicated that the reduction of mRNAs encoding components of the translation machinery was also reflected on the protein level, which could be confirmed by Gene Ontology analysis of affected proteins, demonstrating an enrichment of proteins involved in mRNA processing and splicing (Fig. 4g and Supplementary Data 7).

Taken together, our results show that CBP/p300 inhibition has a pronounced impact on protein synthesis that is at least in part a

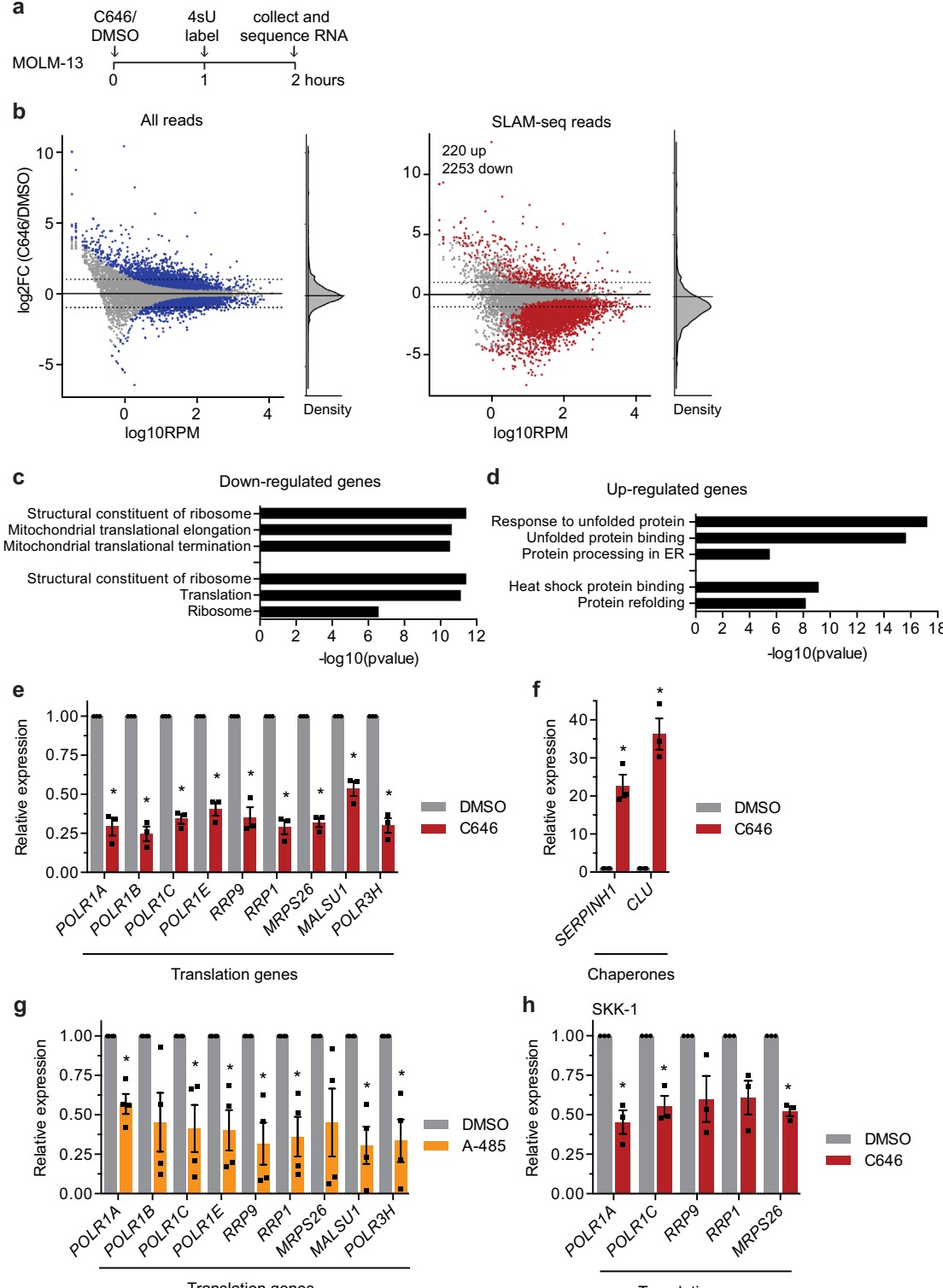

**Fig. 3 Immediate target genes of CBP/p300 identified by SLAM-seq. a** Schematic workflow of the SLAM-seq sample preparation protocol for the metabolic labelling of newly synthesized RNAs. MOLM-13 cells were treated with 10 μM C646 or DMSO, labelled with 4sU after 1 h, the RNA was collected after 2 h and subsequently sequenced. **b** Plots show the fold change (FC) in the abundance of total mRNA or newly synthesized mRNA in relation with their normalized baseline expression in reads per million (RPM). Genes significantly changed (FDR < 0.01) are highlighted in blue and red, respectively. **c** Top enrichment clusters of downregulated or **d** upregulated transcripts as determined by functional annotations clustering using DAVID[68,69]. **e**, **f** Relative expression of selected CBP/p300 target genes by RT-qPCR in MOLM-13 cells treated for 4 h with 10 μM C646 or 0.1% DMSO. **g** Relative expression of selected CBP/p300 target genes by RT-qPCR in MOLM-13 cells treated for 4 h with 1 μM A-485 or 0.01% DMSO. **h** Relative expression of selected CBP/ p300 target genes by RT-qPCR in SKK-1 cells treated for 4 h with 10 μM C646 or 0.1% DMSO. **c**, **d** Statistical analysis was performed by Fisher's exact test. **e–h** Data represent the mean ± SEM of at least three independent experiments. Statistical analysis was performed by two-sided Student's *T*-test compared to untreated samples, *p-value < 0.05. Source data are provided as a Source Data file and in Supplementary Data 4 and 5.

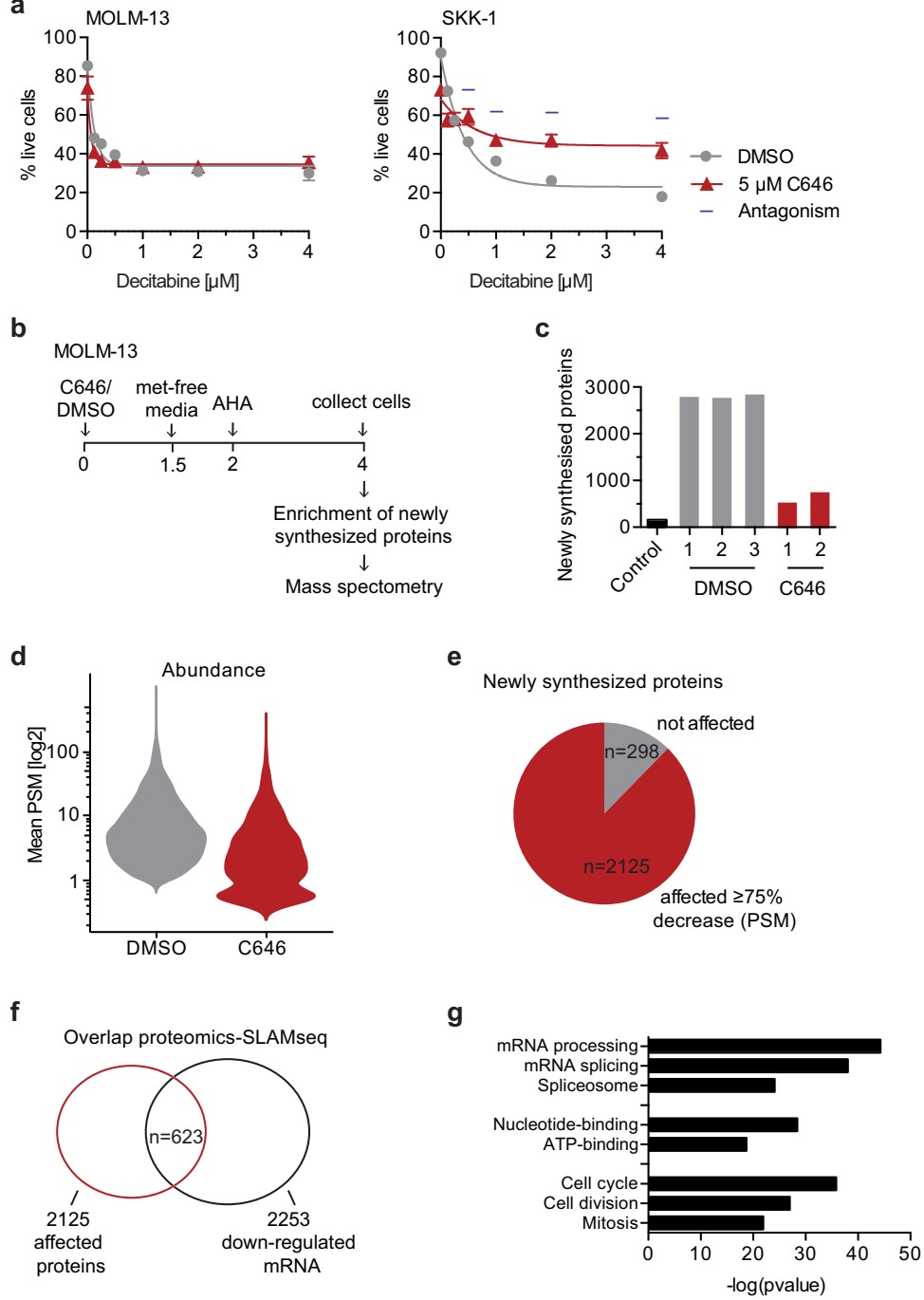

**Fig. 4 CBP/p300 inhibition reduces protein synthesis but does not synergize with decitabine. a** Percentage of live MOLM-13 or SKK-1 cells after 4 days of treatment with indicated concentrations of AZA in combination with 5 μM decitabine or DMSO (0.05%) as vehicle control. Data represent the mean ± SEM of four independent experiments. Statistical analysis was performed using two-way ANOVA. Calculating the combination index (CI) indicated antagonism (CI > 1). **b** Schematic workflow of the detection of nascent proteins. MOLM-13 cells were treated with 10 μM C646 or 0.1% DMSO, the media was changed to methionine (met)-free media after 1.5 h, metabolic AHA label was added for 2 h and cells were collected after a total of 4 h of inhibitor treatment. Newly synthesized proteins were affinity purified following the ClickIT protocol. **c** The number of detected newly synthesized proteins in untreated and treated samples is plotted. An unlabelled sample served as negative control. **d** Violin plot presenting the mean peptide spectrum matches (PSM) of all detected proteins. **e** Pie chart indicating that the majority of detected proteins was affected as defined by a reduction in PSM of 75% or more. **f** Overlap between significantly decreased newly synthesized proteins (≥75% reduction in PSM) determined by ClickIT enrichment and significantly downregulated nascent mRNA (log2FC ≤ −1, FDR ≤ 0.1) determined by SLAM-seq. **g** Ontology analysis of significantly reduced proteins (≤75% reduction compared to untreated samples). Statistical analysis was performed by Fisher's exact test. Source data are provided as a Source Data file and in Supplementary Data 6 and 7.

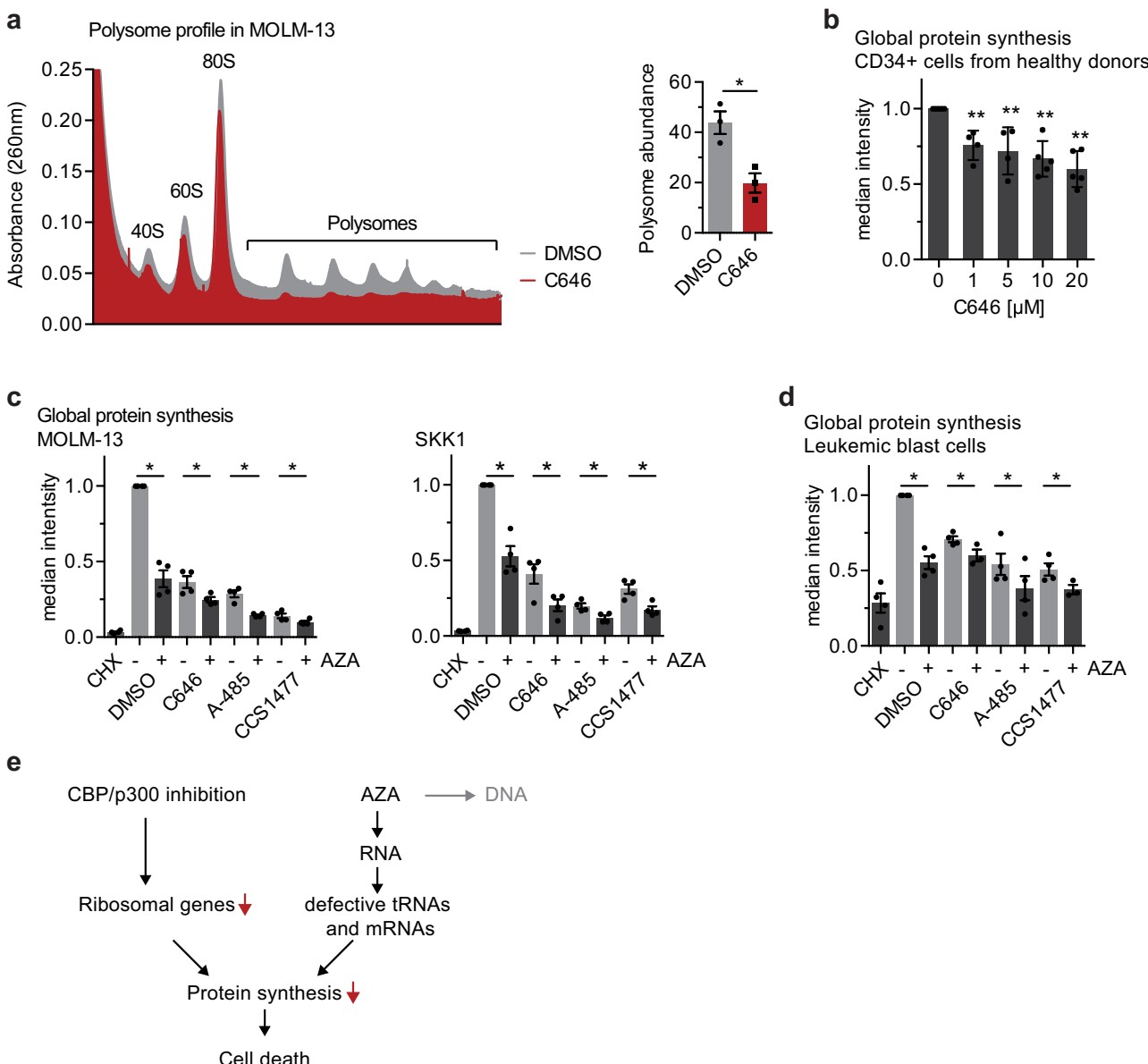

**Fig. 5 CBP/p300 inhibition impairs active protein translation, which can be further reduced by azacitidine. a** Representative polysome profile in MOLM-13 treated with 10 µM C646 for 4 h. Polysomal lysates were fractionated by ultracentrifugation on sucrose gradients and the ribosomal RNA content was monitored across the gradient by UV absorbance (260 nm). The polysome abundance was normalized to the baseline absorbance. Data represent the mean ± SEM of three independent experiments. Statistical analysis was performed by two-sided Student's $T$-test compared to untreated samples, *$p$-value = 0.0267. **b** Median fluorescence intensity of CD34+ cells from healthy donors after 8 h of treatment with increasing concentrations of C646 followed by Protein Synthesis Assay. Data represent the mean ± SEM of five independent experiments. Statistical analysis was performed by unpaired two-sided $T$-test compared to untreated samples, **$p$-value < 0.001. **c** Median fluorescence intensity of MOLM-13 or SKK-1 cells treated for 6 h with DMSO (0.1%), C646 (10 µM), A-485 (1 µM or 2.5 µM) or CCS1477 (1 µM or 2.5 µM) in the absence or presence of AZA (2 or 1 µM) followed by Click-iT™ HPG Alexa Fluor™ 594 assay. Cells treated 1.5 h with cycloheximide (CHX) were used as positive control. **d** Representative images of median fluorescence intensity of leukemic blast cells from AML patients treated for 3 h with the indicated concentrations of DMSO (0.075%), C646 (7.5 µM), A-485 (1 µM) or CCS1477 (0.25 µM) in the absence or presence of 0.25 µM AZA followed by Click-iT™ HPG Alexa Fluor™ 594 assay. Cells treated 1.5 h with cycloheximide (CHX) were used as positive control. **e** Mechanistic model suggesting that the synergy between CBP/p300 inhibition and AZA is caused by the reduction of protein synthesis. **c**, **d** Data represent the mean ± SEM of at least three independent experiments. Statistical analysis was performed by two-sided Student's $T$-test, *$p$-value < 0.05. Source data are provided as a Source Data file and in Supplementary Data 8.

consequence of transcriptional downregulation of target genes encoding components of the translation machinery.

**CBP/p300 inhibition reduces active protein translation and enhances the RNA-dependent functions of AZA.** In order to further explore the effects of CBP/p300 inhibition on protein synthesis, we assessed by sucrose gradient ultracentrifugation the

cellular polysomal content as readout for the activity of the cellular transcriptome, which represents the protein synthetic capacity of the cell[51]. As shown in Fig. 5a (Supplementary Data 8), treatment of MOLM-13 cells with the CBP/p300 inhibitor C646 strongly decreased the amount of poly-ribosomes compared to untreated cells, and indicates a severe impact of C646 on global protein translation. To investigate this further, we

performed flow cytometric assays to globally monitor protein synthesis. We were able to confirm the reduction of protein synthesis in response to CBP/p300 inhibition in primary hematopoietic stem and progenitor cells isolated from the bone marrow of five healthy donors (Donor details in Supplementary Data 2). As shown in Fig. 5b, the treatment of these primary cells with C646 lead to a dose-dependent partial reduction in protein synthesis. In MOLM-13 and SKK-1 cells, the impact on protein synthesis was much more pronounced after treatment with any of the three CBP/p300 inhibitors (Fig. 5c). A less pronounced reduction was visible in the cells stably depleted of CBP (Supplementary Fig. S5). Furthermore, we were able to confirm the inhibitory effect of AZA on translation (Fig. 5c). The combination of AZA with the CBP/p300 inhibitors C646, A-485 or CCS1477 resulted in a significant further decrease in protein synthesis, in some cases approaching the levels of a total shutdown achieved with cycloheximide (CHX, Fig. 5c). Similar to sAML cell lines, leukemic blast cells isolated from the bone marrow of AML patients displayed a strong sensitivity to CBP/p300 inhibitors on the level of protein synthesis, which was further increased in the presence of AZA (Fig. 5d).

Taken together, CBP/p300 inhibition has a pronounced inhibitory influence on protein synthesis. Based on our data, we propose that this mechanism of action of CBP/p300 inhibitors explains their synergy with the RNA-dependent effects of AZA (Fig. 5e).

## Discussion

In our study, we examined the influence of chromatin regulators on the cell-intrinsic response to AZA by performing a genetic loss-of-function screen in an MDS-derived sAML cell line. Using this unbiased approach, we identified the *N*-α-acetyltransferase NAA15 and the transcriptional coactivator CBP as a major regulator of AZA resistance and sensitivity, respectively. NAA15 is an N-terminal α-amino-acetylase that, as part of the NatA complex, promotes sensitivity to apoptotic stimuli by acetylation of key apoptotic mediators[52]. This could explain the resistance to AZA treatment observed in the NAA15 knockdown cells. In contrast, the effect of CBP on AZA sensitivity was unexpected but confirmed by pharmacologic inhibition of CBP and its closely related protein p300 in several AML cell lines. Our approach was limited to the cell-intrinsic response and has focused on cell survival as a primary parameter to assess drug response. It is important to keep in mind that our approach has modelled only one aspect of the mechanism underlying the function of AZA in the clinic. Future studies will need to evaluate the validity of CBP/p300 as a combinatorial drug target in mouse models and co-culture experiments that better represent the complexity of MDS and sAML. This is particularly important, as part of the clinical efficacy of AZA has been attributed to activation of an interferon response through viral mimicry, making disease cells visible to the immune system[18]. If further validated in pre-clinical studies, the availability of biosafe inhibitors of CBP/p300 will warrant their combination with AZA to be tested in a clinical trial assessing improvements of the current therapy of high-risk MDS.

Importantly, here we show that the synergistic effect of CBP/p300 inhibition is specific for AZA and not observed in combination with the closely related drug decitabine. The main difference between both drugs is that 80%–90% of AZA is incorporated into RNA and only the remaining 10%–20% into DNA, whereas decitabine is exclusively incorporated into DNA[14]. A number of previous studies have reported differences in the cellular response to decitabine and AZA[22,26,53], and this included the finding that AZA, but not decitabine, reduces protein synthesis in AML cell lines[22]. As we did not observe a synergy

between CBP/p300 inhibition and decitabine, our results suggest that CBP/p300 inhibition synergized with the RNA-dependent effects of AZA. Both AZA and decitabine are approved for the treatment of haematological diseases including high-risk MDS. Currently, there is no biomarker or patient stratification for choosing one treatment over the other. Although there is no randomized clinical trial that has directly compared both drugs in the same cohort of patients, several studies suggested that the overall response of patients to decitabine and AZA is highly similar[9,54–57]. The partially different mechanisms of action might, however, become relevant when considering response predicting parameters, combinations with other drugs or mechanisms of resistance. Our study points at a beneficial use of CBP inhibitors primarily in combination with AZA and not decitabine. Others have recently shown that resistance to AZA was associated with changes in RNA-dependent chromatin structures and an increased interaction of RNA Polymerase 2 with NSUN1 and Brd4, and loss of interactions with NSUN3 and DNMT2[25]. Importantly, these changes have been attributed to the RNA-dependent functions of AZA and create a vulnerability of AZA-resistant cells for Brd4 inhibition that would not be expected for decitabine-resistant cells. Revisiting our screening results, the knockdowns of NSUN3 and Brd4 had also opposing effects albeit much less pronounced than those observed by knockdown of CBP or NAA15 (see Supplementary Data 3). In support of the study by Cheng at al.[25], we found that knockdown of NSUN3 increased AZA sensitivity.

An unexpected and important finding of our study is that CBP/p300 inhibition affects protein synthesis. As a possible explanation, we found that CBP/p300 positively regulates several genes encoding core components of the translational machinery, and that CBP/p300 inhibition drastically reduced global protein synthesis. Genes encoding chaperones dealing with unfolded proteins were upregulated in response to CBP/p300 inhibition, most likely as a stress response to problems in protein translation. However, based on the fast and pronounced impact of CBP/p300 inhibition on protein synthesis, we cannot exclude the possibility of an additional direct mechanism. Taken together, our results suggest that CBP/p300 inhibition potentiates the RNA-dependent functions of AZA. The convergence of AZA and CBP/p300 inhibition on reducing protein synthesis is a likely explanation for the observed synergy. Here we have primarily used the tool compound C646, which was the first selective inhibitor of CBP/p300 to be developed[42]. However, it lacks potency and its pharmacokinetic properties are unfavourable for use in vivo[44]. To overcome these problems, new inhibitors of CBP/p300 with improved characteristics have been generated. A-485 is such an inhibitor and, similar to C646, inhibits the catalytic domain[44]. Another promising CBP/p300 inhibitor is CCS1477 from Cell-Centric, Ltd[46,58], which, in contrast to C646 and A-485, targets the conserved bromodomain of p300 and CBP. CCS1477 is currently being evaluated first in human clinical trials in AML, multiple myeloma and non-Hodgkin lymphoma (NCT04068597), and in castration-resistant prostate cancer (NCT03568656). Our results demonstrate that irrespective of the mode of inhibition, all tested CBP/p300 inhibitors synergized with AZA and induced a strong reduction of protein synthesis.

The insight that CBP/p300 inhibition affects translation opens an avenue for the rational design of combinatorial treatments. In future studies, we will test potential synthetic lethal combinations of CBP/p300 inhibitors with drugs such as the eIF2 translation factor inhibitor salubrinal. Furthermore, our data suggest that CBP or p300-deficient and haplo-sufficient cells might be particularly sensitive to AZA or treatments directly limiting protein synthesis. This is relevant, as loss of p300 is associated with disease progression[59].

In conclusion, our study provides important insight into drugs relevant for haematological diseases. CBP/p300 inhibition synergistically enhances the cell-intrinsic response to AZA, which is currently used to treat high-risk MDS patients. The synergy is specific for AZA and its RNA-dependent functions, and is not observed with the alternative FDA-approved drug decitabine. CBP/p300 inhibitors have entered clinical trials and the here-recognized link to protein synthesis is informative for the development of potent drug combinations and response predictors.

## Methods

**Plasmids including hEPI9 library**. The cSGEP vector was generated using standard cloning techniques to insert the described UCO element[40] into the previously described mir-E shRNA expression vector SGEP (Addgene #111170)[38]. The shRNA library hEPI9 consisting of 7296 shRNA targeting 912 different chromatin genes (8 shRNAs per target) (Supplementary Data 1) was generated by cloning a PCR-amplified oligo nucleotide pool (Agilent Technologies, Palo Alto, CA) into SGEP and cSGEP following previously described procedures[60]. Individual shRNAs including positive and negative controls (Supplementary Data 2) were inserted into cSGEP following the described cloning procedure[39]. For the production of lenti-viral particles, we used packaging plasmids psPax2 (Addgene #12260) and pCMV-VSV-G (Addgene #8454).

**Cell culture and drug treatments**. MOLM-13 (DMSZ # ACC554), F36-P (DMSZ # ACC543) and SKK-1 have been obtained from DSMZ as a collaboration with Hans Drexler and have been characterized in detail[36]. SKK-1 cells are available from the original authors[37]. MDS-L cells were kindly provided by Tohyama and colleagues[61].

Cells were maintained in RPMI 1640 (Gibco, Thermo Scientific, Waltham, MA) supplemented with 10% of fetal bovine serum (FBS), 1% penicillin–streptomycin and 1% L-Gutamine (Gibco) at 37 °C in 5% $CO_2$. HEK293T (ATCC #CRL-11268) and HS-5 (ATCC #CRL-11882) were obtained from American Type Culture Collection and cultured in Dulbecco's modified Eagle's media (Gibco) supplemented with 10% FBS, 1% penicillin–streptomycin and 1% L-Glutamine (Gibco) at 37 °C in 5% $CO_2$. Cells were authenticated and passaged for <6 months. CD34+ cells were isolated from mononuclear cells of healthy donors (see Supplementary Data 2 for donor characteristics) and cultured in suspension as described[62]. For combinatorial treatments, cells were treated with AZA (Sigma-Aldrich, St. Louis, MO), decitabine (Sigma-Aldrich), C646 (Sigma-Aldrich), A-485 (Tocris, Bristol, UK), iCBP-112 (Tocris), CCS1477 (CellCentric, Ltd, Cambridge, UK), JQ-1 (Sigma-Aldrich), DMSO (Sigma-Aldrich) or CHX (Sigma-Aldrich) at the indicated amounts.

**Viral transduction and generation of stable cell lines**. SKK-1 or MOLM-13 cells were transduced using standard lentiviral transduction procedures. In brief, viral supernatants generated in transfected HEK293T cells were mixed with 8 μg/ml polybrene (Sigma-Aldrich) and used to infect $2 \times 10^6$ cells/well in six-well plates by centrifuging at $500 \times g$ for 30 min at 37 °C. This procedure was repeated once before starting selection with 1 μg/ml puromycin 48 h after the first infection.

**Antibodies and protein analysis**. SKK-1 cells were lysed in RIPA lysis buffer containing protease and phosphatase inhibitors. Protein quantification was done using the BCA Protein Assay Kit (Thermo Scientific) and after boiling in Laemli buffer, equal protein concentrations were loaded and separated on SDS-polyacrylamide gel electrophoresis prior to transfer overnight to nitrocellulose membranes. After blocking with non-fat milk, membranes were incubated with primary antibodies and then with appropriate secondary fluorophore-conjugated secondary antibodies (LI-COR Biosciences, IRDye, 1 : 20,000). The dried membranes were scanned with an Odyssey CLx Imager. Protein levels were quantified using the Odyssey and the band density normalized using the density of the histone H3 signal. The following antibodies were used: anti-CREBBP (Abcam, Cambridge, UK, ab2832, 1 : 1000 dilution) and anti-histone H3 (Abcam, ab1791, 1 : 10,000 dilution).

**Real-time quantitative PCR**. For RT-qPCR, RNA was extracted using the Max-well® RSC simplyRNA Cells Kit (Promega, Madison, Wisconsin) and reverse transcribed with the First strand cDNA synthesis kit (Thermo Scientific). The cDNA was PCR amplified in triplicate using the Fast SYBR green dye on the Applied Biosystems™ QuantStudio™ 7 Flex Real-Time PCR System. Relative expression was determined using parental cells or cells expressing the control cSGEP vector as reference samples and *GAPDH* as the internal control. Sequences of primers are provided in Supplementary Data 2. Statistical analysis was performed by two-sided Student's *T*-test using the GraphPad Prism software (version 6).

**Loss-of-function screen**. During the whole experiment, we aimed to have a 1000× representation of the hEpi9 library containing 7296 different plasmids. Taking into account the infection efficiency of 10% determined prior to the experiment to achieve single-copy integration, $7.3 \times 10^7$ cells were transduced in triplicates as described above. After puromycin treatment, each replicate was split in two and one half treated with 0.25 μM AZA every 2 days. During the treatment, cells were kept at a density of ~$0.5 \times 10^6$ cells/ml, ensuring that the cell numbers never dropped under $7.3 \times 10^6$ to preserve library representation. Fourteen days after initiation of the treatment, the cells were collected and prepared for sequencing as previously described[39]. The analysis of the sequencing results was performed using the R package EdgeR (R Core team 2019) as described[63]. Gene-level analysis was performed with the function Roast[64] and the results ranked on a gene-by-gene level (Supplementary Data 3).

**Flow cytometric analysis of viability and apoptosis**. Following 4 days incubation with the indicated inhibitors, cell viability was assessed by flow cytometry of cells stained with 1 μg/mL DAPI (4′,6-Diamidino-2-phenylindole dihydrochloride) (Thermo Scientific), and 100 μM MitoTracker® Red CMXRos (Thermo Scientific), using the LSR Fortessa cytometer and the BD FACSDiva™ software (BD Biosciences, Franklin Lakes, New Jersey) (see gating strategy in Supplementary Fig. S6). Statistical analysis (analysis of variance test) and IC50 values were calculated using GraphPad Prism software (version 6). Chou–Talalay's CI method using the CompuSyn software (version 1) was used to calculate the additive effect (CI = 1), synergy (CI < 1) or antagonism (CI > 1) in the drug combinations[65]. The percentage of apoptotic cells was determined by flow cytometry of cells stained with Annexin V-APC (BD Biosciences, 1 : 200 dilution) and 1 μg/mL DAPI.

**Identification of direct target genes using SLAM-seq analysis**. Next, $5 \times 10^6$ MOLM-13 cells were either treated with 10 μM C646 or DMSO (three replicates each) for 1 h followed by labelling with 100 μM 4sU for another hour. RNA was extracted using the RNeasy Mini kit (Qiagen, Hilden, Germany) and SLAM-seq was performed using the SLAM-seq Anabolic Kinetics Kit (Lexogen Gmbh, Austria) followed by library preparation using the QuantSeq 3′mRNA-Seq for Illumina (FWD) and PCR Add-on Kit for Illumina (Lexogen Gmbh, Austria) according to the manufacturer's instructions. The subsequent data analysis has essentially been performed as described in Muhar et al.[50] using the software SLAM-Dunk v0.4.1[66] (http://t-neumann.github.io/slamdunk/index.html) and the R package DEseq2[67].

In brief, the reads were aligned to the human genome (GRCh38) and annotated to the 3′-untranslated region (UTR) reference using the slamdunk all function with default settings[50]. For the differential expression analysis, reads mapping to different UTR annotations of the same genes were collapsed and differential expression called on SLAM-seq reads with ≥2 T > C conversions using the R package DESeq2 version 1.26.0[67] (Supplementary Data 4). For comparison, differential expression analysis was also performed on all reads (Supplementary Data 4). Gene Ontology analysis was performed on the most significant deregulated genes (FDR ≤ 0.01, log2FC ≤ −1 or ≥1) using DAVID 6.8[68,69] (Supplementary Data 5).

**Analysis of de novo protein synthesis**. De novo protein synthesis was assessed using the ClickIT Enrichment Kit (Thermo Scientific) and ClickIT-AHA (L-Azidohomoalanine) (Thermo Scientific) according to the ClickIT® Protein Enrichment Kit instructions. In brief, $10 \times 10^6$ MOLM-13 cells were treated with 10 μM C646 for 2 h or 0.1% DMSO, AHA labelled (in methionine-free media) for another 2 h and subsequently processed using the indicated kits. Liquid chromatography-MS/MS was performed employing an Advion TriVersa NanoMate (Advion, Ithaca, NY) fitted on an Orbitrap Fusion Lumos™ Tribrid mass spectrometer (Thermo Scientific). The data were acquired with Xcalibur software vs. 4.0.27.10 (Thermo Scientific) and analysed using the MaxQuant software (1.6.7.0). The data were aligned to the Swissprot/Uniprot human database downloaded from the www.uniprot.org website (October 2019). A target and decoy database were used to assess the FDR, which was set to 1% at both peptide and protein level. Trypsin was chosen as an enzyme and a maximum of two missed cleavages were allowed. Carbamidomethylation (c) was set as a fixed modification, whereas oxidation (M) and acetylation (N-terminal) were used as variable modifications. Searches were performed using a peptide tolerance of 7 p.p.m. and a product ion tolerance of 0.5 Da.

Proteins were filtered by removing 'Potential Contaminant', 'Reverse' and 'Identified Only by Site', background proteins appearing in the negative control (cells non-labelled with AHA), as well as proteins with PSM = 0 in one or more DMSO control samples. Finally, proteins disappearing due to the treatment and the proteins with a decrease of >75% of the number of PSMs in the treated cells compared to DMSO control were included in the final list of candidates (Supplementary Data 6). Gene Ontology analysis of genes corresponding to the significantly affected proteins (≥75% reduction compared to DMSO control) was performed using DAVID 6.8[68,69] (Supplementary Data 7).

**Bone marrow samples**. All experiments and methods were performed in accordance with relevant guidelines and regulations. Specifically, experiments using

human leukemic blast cells from AML patients were approved by the Ethics committee of the Hospital Germans Trias i Pujol (Badalona, Spain; CEIm IRB00002131) for the protocol RESPONSE (PIE16/00011). Bone marrow samples from healthy donors were obtained from femoral heads of patients undergoing hip replacement surgery and the experiments approved by the Ethics commission of the Faculty of Medicine (Technical University of Munich, Germany). Written informed consent in accordance with the Declaration of Helsinki was obtained from all patients.

Bone marrow samples from AML patients (see Supplementary Data 2 for patient details) have been obtained from the IJC-Campus ICO-GTP Biological Sample Collection. Only samples from patients with high blast count (>75) were used. Samples were thawed, seeded on top of HS-5 stroma cells and a few hours later treated with different concentrations of AZA. One day later, cells were treated with different concentrations of A-485, CCS1477 or DMSO for two additional days. The number of apoptotic cells was determined using Annexin V-APC as described above. To exclude the signal from co-cultured HS-5 cells, samples were co-stained with Pharmingen™ PE Mouse Anti-Human CD90 Clone: 5E10 (BD Biosciences, Catalogue number: 561970, 1 : 400 dilution) and only the CD90-negative fraction used for further analysis (see gating strategy in Supplementary Fig. S6).

CD34+ cells were isolated from healthy donors (see Supplementary Data 2 for patient details) with immunomagnetic beads (Miltenyi Biotec, Bergisch Gladbach, Germany) and cultured for 48 h.

**Flow cytometric analysis of protein synthesis.** CD34+ cells were washed and resuspended in media with different concentrations of C646 (Sigma-Aldrich) and incubated for another 8 h after which protein synthesis was analysed with the Protein Synthesis Assay Kit Red (Abcam) according to the manufacturer's instructions. The samples were analysed on a Beckman Coulter Cyan ADP (Beckman Coulter) and with FlowJo v10 (FlowJo LLC). The median fluorescence intensity of each condition was normalized to the DMSO control. Statistical analysis (unpaired $t$-test) were calculated using GraphPad Prism software (version 6).

Leukemic blast cells or SKK-1 and MOLM-13 cells were treated with different concentrations of AZA, C646, A-485, CCS1477 or DMSO and incubated for 3 or 6 h, respectively. Cells treated 1.5 h with CHX were used as a positive control. The samples were then processed using the Click-iT™ HPG Alexa Fluor™ 594 kit (Thermo Scientific) according to the manufacturer's instructions and the protein synthesis analysed using the LSR Fortessa cytometer (BD Biosciences) and FlowLogic v8 (Inivai Technologies, Victoria, Australia).

**Polysome profiling.** Analysis of distribution of RNA across sucrose gradients was performed as described earlier[70], except for minor modifications. Briefly, $5 \times 10^6$ MOLM-13 cells were treated with 10 μM C646 or DMSO and incubated for 4 h. Equal amounts of polysomal lysate (170–500 μg) were loaded on 10–50% sucrose linear gradients generated with a BIOCOMP gradient master and containing 80 mM NaCl, 5 mM MgCl₂, 20 mM Tris HCl pH 7.4, 1 mM dithiothreitol, 10 U/ml Rnase inhibitor. Gradients were centrifuged on a SW40Ti rotor for 2 h 30 min at $218,000 \times g$ and analysed on a BIOCOMP gradient station. Quantification of polysomal absorbance at 260 nm was normalized to the baseline absorbance.

**LINE-1 methylation assay.** Response to decitabine was validated using the Global DNA Methylation LINE-1 kit (Active Motif, Carlsbad, CA). For this, MOLM-13 cells were treated with 1 μM decitabine or DMSO and 24 h later the genomic DNA collected and processed according to the manufacturer's instructions.

**Statistical analysis.** Statistical analyses were performed using the GraphPad Prism software (version 6) and suitable tests as indicated in legends.

**Reporting summary.** Further information on research design is available in the Nature Research Reporting Summary linked to this article.

## Data availability

The data that support this study are available from the corresponding author upon reasonable request. The proteomics data generated in this study have been deposited in the PRIDE database under accession code PXD021738. The shRNA screen data generated in this study have been deposited in the GEO database under accession code GSE158738. The SLAM-Seq data generated in this study have been deposited in the GEO database under accession code GSE158813. The protein reference used in this study is available in the Swissprot/Uniprot human database (www.uniprot.org website, October 2019). The human genome reference GRCh38 used in this study is available at https://www.ncbi.nlm.nih.gov/. Source data are provided with this paper.

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

## Acknowledgements

This project was supported by the FEDER/Ministerio de Ciencia e Innovación - Agencia Estatal de Investigación through the grant PIE16/00011 RESPONSE (to M.B.), the European Research Council ERC-StG-336860 (to J.Z.), the grant Juan de la Cierva—Formación FJCI-2014-22983 (to J.D.), the grant Sara Borrell CD17/00084 (to J.D.), the Marie Skłodowska Curie Training network "ChroMe" H2020-MSCA-ITN-2015-675610 (to M.M.), the FPI predoctoral fellowship BES-2016–077251 (to M.M.L.P.), the SFB 1243 DFG (to K.S.G.), the Austrian Science Fund SFB-F4710 (to J.Z.) and the Deutsche José Carreras Leukämie Stiftung DJCLS 14R/2018 (to K.G. and M.B.). Research in the Buschbeck lab is further supported by the following grants: MINECO grant RTI2018-094005-B-I00 (to M.B.), the Marie Skłodowska Curie Training network 'INTERCEPT-MDS' H2020-MSCA-ITN-2015-953407 (to M.B. and K.S.G.); AGAUR 2017-SGR-305 (to M.B.) and Fundació La Marató de TV3 257/C/2019 (to M.B.). Research at IMP is supported by Boehringer Ingelheim, the Austrian Research Promotion Agency (Headquarter grant FFG-852936) and the Austrian Academy of Sciences. Research at the IJC is generously supported by the 'La Caixa' Foundation, the Fundació Internacional Josep Carreras, Celgene Spain and the CERCA Programme/Generalitat de Catalunya. A.G. received funds by "Agencia Estatal de Investigación" (AEI) through the Plan Nacional "Excelencia" grant number SAF2017-84301-P, by the "Associación Española Contra el Cancer" (AECC) grant number LABAE20040GENT and by the Agency for Management of University and Research Grants (AGAUR) of the Catalan Government grant 2017SGR01743. Proteomic analyses were performed in the IJC Proteomic Unit, which are part of Proteored PRB3 and are supported by grant PT17/0019 from the PE I + D + i 2013-2016, funded by ISCIII and ERDF. We thank Kaoru Tohyama for providing MDS-L cells, members of the Buschbeck lab, the RESPONSE network (PIE16/00011), Blanca Xicoy and Francesc Solé for valuable discussions. We thank the IJC Biobanking unit for sample preparation and storage, Bernat Cucurull from the IJC Proteomics unit for the ClickIT sample preparation, Marco Fernandez for advice and training in flow cytometry and all other staff of IJC and IGTP core facilities for excellent support.

## Author contributions

Conceived and designed the experiments: J.D. and M.B. Designed and generated essential tools: M.F., P.R., J.Z. and N.B. Performed the experiments: J.D., R.W., M.M., M.M.L.P., R.C., M.V.D.G., C.D.L.T., C.M.H. and A.G. Analysed the data: J.D., M.M.L.P., C.D.L.T., J.B., C.M.H. and A.G. Contributed reagents/materials/analysis tools: M.a.M., K.S.G., L.Z., J.Z. and N.B. Wrote the paper: J.D. and M.B.

## Competing interests

K.S.G. has received support for independent research from Celgene Corp. N.B. is an employee and stockholder in CellCentric, Ltd. The other authors declare no competing interests.
