## [Peer Review File · Nature Communications]

REVIEWER COMMENTS

Reviewer #1 (Remarks to the Author):

5-azacytidine (AZA) is a paradigmatic DNA methyltransferase inhibitor that is approved for the treatment of myelodysplastic syndromes. However, the mode of action of the drug remains poorly understood. In this paper, Diesch et. al. claim CBP as a major regulator of AZA sensitivity and then claim an effect of CBP on protein synthesis. This is an interesting and potentially impactful study, as it addresses underinvestigated aspects of AZA. Generally, the paper is well written and the data are clearly presented. However, key claims are currently not supported by experimental data and need to be substantiated. Most importantly:

1. It needs to be confirmed that CBP inhibition limits global protein synthesis. The protein synthesis assay that was used for Fig. 5H needs to be used for analyzing protein synthesis in CBP-depleted and inhibitor-treated (AZA, C646, both) SKK-1 and MOLM-13 cells. In addition, protein synthesis needs to be analyzed by polysome profiling.
2. The synergistic interaction between AZA and CBP inhibitors needs to be confirmed in primary cells from multiple independent donors (Table S2).

Additional points:

3. The specificity of the C646 should be confirmed by demonstrating that there are no additive effects of C646 in CBP knockdown cells.
4. Line 208: why do the authors consider an $FDR < 0.1$ a stringent cut-off? How would the numbers look with an actual stringent cut-off ($FDR < 0.01$)?
5. Please specify whether Fig. 4C and D show top hits from the pathway analysis.
6. Literature findings on the effects of AZA on RNA and protein synthesis often appear superficially described and overstated. The authors should be more careful and more detailed when describing the published material. What was actually shown in these papers?

7. A mechanistic model for the interaction between AZA and CBP/p300 and the effect of the latter on protein synthesis needs to be developed.

Reviewer #2 (Remarks to the Author):

In this interesting report by Diesch et al, the authors report on their findings using a loss of function shRNA screen, that the histone acetyl transferase CBP is a major regulator of azacitidine sensitivity and that compounds inhibiting the enzymatic activity of CBP/p300 synergize with azacitidine in reducing viability of AML cell lines. These synergistic effects were specific to azacitidine which is predominantly incorporated into RNA, and not observed with the deoxycytidine analogue decitabine

Specific comments:

1) A major thrust of this paper are the translational implications of this work with regard to uncovering a novel regulator of azacitidine sensitivity and the suggestion of combinatorial approaches that could potentially improve the clinical activity of azacitidine. Therefore, investigating this in other more relevant model systems- for example PDX models or primary AML cells would be important to affirm these findings and confirm the translational relevance

2) In discussing the RNA mediated effects of azacitidine which have been traditionally understudied, it would be important to discuss the work of Cheng J et al, Nature Communications 2018; 9 (1); 1163, with regard to RNA cytosine methylation and RNA cytosine methyl transferases (RCMT) mediating chromatin organization and azacitidine response and resistance in leukemia. An important aspect of that work was the demonstration that the RCMT- NSUN1 binds BRD4 and RNA-polymerase-II to form an active chromatin structure that is resistant to azacitidine, but sensitive to the BRD4 inhibitor JQ1. The authors in that paper also demonstrated this chromatin structure occurs in primary leukemia cells derived from patients that have shown clinical resistance to azacitidine.

Given the parallels of the work by Cheng et al with the current work, it is important for the authors to discuss how the current work fits with/contributes to/ is distinguished from previous work done in this regard.

3) The authors imply in the discussion that azacitidine is associated with improved survival (relative to decitabine) in a prior study "in a direct comparison of both treatment schemes." In reality, there are no randomized studies comparing azacitidine use (head to head) with decitabine use in the same patient population-high risk MDS/AML- within the same clinical trial. Making comparisons across

clinical trials is fraught with the issue of patient heterogeneity making such analyses inconclusive. Analysis utilizing SEER based population data comparing azacitidine and decitabine use in higher risk MDS patients has shown absolutely no difference in overall survival (p value =1.0)- Zeidan et al, Blood 2018; 131 (7), 818-821

Therefore, although the RNA based mechanism of action is unique to azacitidine relative to decitabine as demonstrated by this work and the work by Cheng et al, it is presently unclear whether one agent is clinically superior to the other. This nuance needs to be appropriately addressed in the discussion

Reviewer #3 (Remarks to the Author):

In their manuscript, "Inhibition of CBP synergizes with the RNA-dependent mechanisms of azacitidine by limiting protein synthesis", Diesch and colleagues describe their identification of CBP/p300 as regulators of sensitivity to azacitidine (AZA) using a loss-of-function shRNA screen. This insight could be an important break-through in our understanding of a major drug used for MDS and AML patients- but several issues could strengthen the work, as outlined below:

Major concerns:

1. In the Abstract, the authors say that it is "surprising" that their findings are specific to AZA and do not extend to decitabine. However, the two drugs are chemically distinct: the nucleoside-like base being attached to ribose in AZA, which causes most of the drug to be incorporated into RNA; in decitabine, the same base is attached to deoxyribose, which directs its incorporation mainly into DNA. In the Introduction, the authors do not really explain the chemical difference between the two hypomethylating agents, but the chemical structure of the drugs completely explains where the drugs get incorporated. The authors may want to add this information to their Introduction. When the authors explain this difference briefly in the Discussion, it comes very late for the reader.

Thus, it seems perfectly logical that the drugs work by quite distinct mechanisms, and we should not be surprised that this group's findings are specific for AZA. The authors may want to adjust their wording in the Abstract and elsewhere to emphasize this.

2. The Figure Legend for Figure 1 does not match the figure. There are 5 elements to Figure 1, A-E, but the legend only goes up to D. The call-outs in the manuscript do not appear to be correct. In Figure 1, the authors highlight CBP, but two other genes seem significant, but the authors do not highlight them-- what are they? Figure 1 would be easier for the reader to follow if the "left" side versus the "right" side of the volcano plot were labeled as to the expected effect on cell growth: Left side of the volcano plot=genes whose products are required for cells survival in AZA; Right side of the volcano plot=genes whose products inhibit cell survival in AZA.

3. For Figure 2, the authors should provide Western blots that correspond to shRNAs- to show knock-down at the protein level.

4. Figure 3 really does not provide new information; the additional cell line and drugs could be combined into Figure 2.

5. The authors should explain why they chose to do SLAM-seq in MOLM-13 cells; and if they obtained similar results with SKK-1 cells. The authors should explain why they seem to switch to using MOLM-13 cells for the remainder of the work presented.

6. In Figure 5, the authors should provide a positive control for treatment by decitabine in the cells (e.g., demethylation of LINE-1 elements). They need to show the readers that their drug treatment results in the expected phenotypes.

7. As the authors may know, we observe tremendous clinical utility in combining AZA with the BCL2 inhibitor venetoclax. What happens with venetoclax is introduced into their system? This is an important experiment. For example, could BCL2 have served as a positive control for their screen?

8. The authors appear to only study two cell lines. Primary MDS cells should be used in validation experiments.

9. The authors should explain/contrast the cytogenetic/molecular mutations found in the SKK-1 and MOLM-13 cells used in the work. The authors cite their initial publication of the SKK-1 cell line, but it would be helpful for the reader if they briefly outlined the key features of these cell lines that make them relevant for these studies. The authors briefly describe these for the MOLM-13 cells, but in an off-handed way.

10. A prior publication [Cheng, J.X. et al. RNA cytosine methylation and methyltransferases mediate chromatin organization and 5-azacytidine response and resistance in leukaemia. *Nat Commun* 21;9(1):1163, 2018. PMID: 29563491 PMCID: PMC5862959] showed that RNA 5-methylcytosine (RNA:m5C) and the RNA:m5C methyltransferases (RCMTs) NSUN3 and DNMT2 directly bind hnRNPk, which interacts with lineage-determining transcription factors (TFs), GATA1 and SPI1/PU.1, and with CDK9/P-TEFb to recruit RNA-polymerase-II at nascent RNA, leading to formation of 5-Azacytidine (5-AZA)-sensitive chromatin structure. The group also showed that NSUN1 binds BRD4 and RNA-polymerase-II to form an active chromatin structure that is insensitive to 5-AZA, but hypersensitive to the BRD4 inhibitor JQ1. AZA-resistant leukemia cell lines and primary MDS/AML samples with AZA-resistance had increases in RNA:m5C and NSUN1-/BRD4-associated active chromatin. How can the authors reconcile their findings with this prior publication? Did their screen identify any of these molecules? [NSUN3 and BRD4 appear DNMT2 in Suppl Table 3, for example] Why is this publication- which seems directly relevant to their model- not cited/discussed?

Minor concerns:

1. "led" is misspelled in line 42.
2. In line 54, the authors should use the adverb "particularly", not the adjective "particular".
3. In lines 57-58, the authors describe the use of hypomethylating drugs as being the treatment of choice for patients not eligible for allogeneic stem cell transplant. However, many patients are bridged to allogeneic stem cell transplant using these drugs, so this sentence does not really reflect how these drugs are used clinically.

REVIEWER COMMENTS

We would like to thank the reviewers for their efforts, their support and their insightful comments. In particular, we were glad to read that reviewer 1 considered our study to be 'interesting' and that reviewers 2 and 3 recognized the 'potential impact' of our findings related to a new drug action and resulting synergies. In particular, we were happy to read that reviewer 3 considered that 'this insight could be an important breakthrough in our understanding of a major drug used for MDS and AML patients'. All reviewers' have raised some concerns and made helpful suggestions of how to further strengthen our manuscript and its key statements. Based on the comments, we have now expanded our results by adding additional experiments and improved the discussion of our results in the context of previous knowledge.

Reviewer #1 (Remarks to the Author):

5-azacytidine (AZA) is an paradigmatic DNA methyltransferase inhibitor that is approved for the treatment of myelodysplastic syndromes. However, the mode of action of the drug remains poorly understood. In this paper, Diesch et. CBP as a major regulator of AZA sensitivity and then claim an effect of CBP on protein synthesis. This is an interesting and potentially impactful study, as it addresses underinvestigated aspects of AZA. Generally, the paper is well written and the data are clearly presented. However, key claims are currently not supported by experimental data and need to be substantiated. Most importantly:

We would like to thank this reviewer for their effort and for considering our study as 'interesting and potentially impactful study'. We fully agree that the RNA function is an 'underinvestigated aspect of AZA'.

1. It needs to be confirmed that CBP inhibition limits global protein synthesis. The protein synthesis assay that was used for Fig. 5H needs to be used for analyzing protein synthesis in CBP-depleted and inhibitor-treated (AZA, C646, both) SKK-1 and MOLM-13 cells. In addition, protein synthesis needs to be analyzed by polysome profiling.

We agree with the reviewer and have performed protein synthesis assays in CBP knockdown cells (Suppl. Fig. S5), treated SKK-1 and MOLM-13 cells (Fig. 5C) as well as treated leukemic blast cells from AML patients (Fig. 5D). As suggested, we have now performed polysome profiling. As shown in Figure 5A, treatment of MOLM-13 cells with C646 strongly reduced the fraction of polysomes. Together these results allow us to strengthen the statement that CBP/p300 inhibition has a pronounced inhibitory impact on global protein synthesis.

2. The synergistic interaction between AZA and CBP inhibitors needs to be confirmed in primary cells from multiple independent donors (Table S2).

We thank the reviewer for this comment and were aware that the reliance on cell lines was a major weakness of our study. I am proud that we have been able to establish first primary AML cultures in our lab. As shown in Figure 2E, we have performed co-treatment experiments in leukemic blast cells isolated from the bone marrow of two AML patients. For this we have used the CBP/p300 inhibitors A-485 and CCS1477 in absence or presence of AZA. While the results are more variable as observed in cell lines, we do still see an increase in apoptosis when using the drug combination compared to AZA mono-treatments. In addition, we have included results from experiments with two additional MDS-derived sAML cell lines (F-36P and MDS-L) in the manuscript that similarly to SKK-1 and MOLM-13 demonstrate additive to synergistic drug interactions (Fig. 2D).

In further support of point 1, we have taken advantage of the primary cultures to confirm the impact of CBP/p300 inhibition on protein synthesis in leukemic blast (Fig. 5D).

Additional points:

3. The specificity of the C646 should be confirmed by demonstrating that there are no additive effects of C646 in CBP knockdown cells.

We have now conducted the experiments as suggested. As shown Suppl. Fig. S3B, we observe a sensitising effect of C646 in the CBP knockdown cells, which added to the effect of the genetic CBP depletion. This is probably due to the facts that the achieved knockdown of CBP was not complete but partial (~50-75% according Suppl. Fig. S1C-D) and that C646 and all other CBP inhibitors also target p300.

4. Line 208: why do the authors consider an $FDR < 0.1$ a stringent cut-off? How would the numbers look with an actual stringent cut-off ($FDR < 0.01$)?

The cut-off was actually $FDR < 0.01$. We apologize for the error in the text and have now corrected this in the main text (page 8) and methods section (page 16).

5. Please specify whether Fig. 4C and D show top hits from the pathway analysis.

Yes, only top hits are illustrated (now Fig. 3C and D). We have clarified this in the figure legend.

6. Literature findings on the effects of AZA on RNA and protein synthesis often appear superficially described and overstated. The authors should be more careful and more detailed when describing the published material. What was actually shown in these papers?

We have carefully revised the available literature and modified the text sections referring to differences between AZA and decitabine in the introduction (page 3, bottom paragraph) and the discussion (page 11, second paragraph).

7. A **mechanistic model** for the interaction between AZA and CBP/p300 and the effect of the latter on protein synthesis needs to be developed.

We have now summarized our results and conclusions in a simplified scheme. This has been added as panel Figure 5E.

Reviewer #2 (Remarks to the Author):

In this interesting report by Diesch et al, the authors report on their findings using a loss of function shRNA screen, that the histone acetyl transferase CBP is a major regulator of azacitidine sensitivity and that compounds inhibiting the enzymatic activity of CBP/p300 synergize with azacitidine in reducing viability of AML cell lines. These synergistic effects were specific to azacitidine which is predominantly incorporated into RNA, and not observed with the deoxycytidine analogue decitabine.

We would like to thank this reviewer for their insightful comments and in particular for providing their valuable clinical point of view that has helped us to improve strength of our results and their discussion in the disease context.

Specific comments:

1) A major thrust of this paper are the translational implications of this work with regard to uncovering a novel regulator of azacitidine sensitivity and the suggestion of combinatorial approaches that could potentially improve the clinical activity of azacitidine. Therefore, investigating this in other more relevant model systems- for example PDX models or primary AML cells would be important to affirm these findings and confirm the translational relevance

We thank the reviewer for this comment and fully agree that the major findings of this study should be validated in more suitable models. My lab is a primarily basic research group and while PDX models are beyond our current capacities, I am proud that we were able to establish stroma cell-supported cultures of primary AML blasts in my lab. As shown in Figure 2E, co-treatments increased the amount of apoptosis in respect to the AZA mono-treatments. As expected these results were more variable than in cell lines. In addition, we have included results from experiments with two additional MDS-derived sAML cell lines (F-36P and MDS-L) in the manuscript that similarly to SKK-1 and MOLM-13 indicated additive to synergistic drug interactions (Fig. 2D). We hope that the insight from our mechanistic research project will encourage experimental haematologists to take the translation of the drug combination further by testing it in more advanced model systems. To increase the potential impact of our study, we have initiated a collaboration with CellCentric Ltd and included their last generation inhibitor CCS1477 in our study. CCS1477 is the first inhibitor of CBP/p300 to be tested in first in human clinical trials in acute myeloid leukaemia, multiple myeloma and Non-Hodgkin lymphoma (NCT04068597) and in castration-resistant prostate cancer (NCT03568656). Representatives of CellCentric pointed out that they are interested to further test the synergy of CCS1477 with AZA in a mouse model.

2) In discussing the RNA mediated effects of azacitidine which have been traditionally understudied, it would be important to discuss the work of Cheng J et al, Nature Communications 2018; 9 (1); 1163, with regard to RNA cytosine methylation and RNA cytosine methyl transferases (RCMT) mediating chromatin organization and azacitidine response and resistance in leukemia. An important aspect of that work was the demonstration that the RCMT- NSUN1 binds BRD4 and RNA-polymerase-II to form an active chromatin structure that is resistant to azacitidine, but sensitive to the BRD4 inhibitor JQ1. The authors in that paper also demonstrated this chromatin structure occurs in primary leukemia cells derived from patients that have shown clinical resistance to azacitidine.

Given the parallels of the work by Cheng et al with the current work, it is important for the authors to discuss how the current work fits with/contributes to/ is distinguished from previous work done in this regard.

We apologize for this omission. This important study had escaped our attention and is now presented in the introduction (page 3, bottom paragraph). We further discuss our results in the context of this study (page 11).

3) The authors imply in the discussion that azacitidine is associated with improved survival (relative to decitabine) in a prior study "in a direct comparison of both treatment schemes." In reality, there are no

randomized studies comparing azacitidine use (head to head) with decitabine use in the same patient population-high risk MDS/AML- within the same clinical trial. Making comparisons across clinical trials is fraught with the issue of patient heterogeneity making such analyses inconclusive. Analysis utilizing SEER based population data comparing azacitidine and decitabine use in higher risk MDS patients has shown absolutely no difference in overall survival (p value =1.0)- Zeidan et al, Blood 2018; 131 (7), 818-821 Therefore, although the RNA based mechanism of action is unique to azacitidine relative to decitabine as demonstrated by this work and the work by Cheng et al, it is presently unclear whether one agent is clinically superior to the other. This nuance needs to be appropriately addressed in the discussion

We would like to thank this reviewer for their comment. After carefully reading the cited study and related works, we fully agree with this reviewer and have modified our discussion accordingly (page 11, second paragraph).

Reviewer #3 (Remarks to the Author):

In their manuscript, "Inhibition of CBP synergizes with the RNA-dependent mechanisms of azacytidine by limiting protein synthesis", Diesch and colleagues describe their identification of CBP/p300 as regulators of sensitivity to azacitidine (AZA) using a loss-of-function shRNA screen. This insight could be an important break-through in our understanding of a major drug used for MDS and AML patients- but several issues could strengthen the work, as outlined below:

We thank this reviewer for their supportive evaluation and for recognizing our results as a potential 'break-through'. We also thank for helpful comments and suggestions that we aimed to address in full as described below.

Major concerns:

1. In the Abstract, the authors say that it is "surprising" that their findings are specific to AZA and do not extend to decitabine. However, the two drugs are chemically distinct: the nucleoside-like base being attached to ribose in AZA, which causes most of the drug to be incorporated into RNA; in decitabine, the same base is attached to deoxyribose, which directs its incorporation mainly into DNA. In the Introduction, the authors do not really explain the chemical difference between the two hypomethylating agents, but the chemical structure of the drugs completely explains where the drugs get incorporated. The authors may want to add this information to their Introduction. When the authors explain this difference briefly in the Discussion, it comes very late for the reader.

Thus, it seems perfectly logical that the drugs work by quite distinct mechanisms, and we should not be surprised that this group's findings are specific for AZA. The authors may want to adjust their wording in the Abstract and elsewhere to emphasize this.

We would like to thank the reviewer for making this important point. We have now removed the word surprising from abstract and attempted to provide a better description of the drugs and their differences in the introduction (page 3, bottom paragraph).

2. The Figure Legend for Figure 1 does not match the figure. There are 5 elements to Figure 1, A-E, but the legend only goes up to D. The call-outs in the manuscript do not appear to be correct.

We apologize for this error and have corrected this now.

In Figure 1, the authors highlight CBP, but two other genes seem significant, but the authors do not highlight them-- what are they?

The two hits for which the shRNAs were also significantly down-regulated are CBX3 and HMGA2. To make them more visible, we have now labelled them in the volcano plot in Fig. 1E.

Figure 1 would be easier for the reader to follow if the "left" side versus the "right" side of the volcano plot were labeled as to the expected effect on cell growth: Left side of the volcano plot=genes whose products are required for cells survival in AZA; Right side of the volcano plot=genes whose products inhibit cell survival in AZA.

We thank the reviewer for this suggestion and have modified the volcano plot accordingly (Fig. 1E).

3. For Figure 2, the authors should provide Western blots that correspond to shRNAs- to show knock-down at the protein level.

Western blots representing the CBP knockdown and their corresponding quantification has been added as Suppl. Fig. S1D.

4. Figure 3 really does not provide new information; the additional cell line and drugs could be combined into Figure 2.

We agree with the reviewer and have reorganized the figures. Figure 1 now also includes the technical validation with individual hairpins in SKK-1 cells, while Figure 2, as suggested, summarizes all data confirming the additive to synergistic interaction between CBP/p300 inhibition with different drugs, additional cell lines and primary AML patient blasts.

5. The authors should explain why they chose to do SLAM-seq in MOLM-13 cells; and if they obtained similar results with SKK-1 cells. The authors should explain why they seem to switch to using MOLM-13 cells for the remainder of the work presented.

Thank you for this comment. We have switched to MOLM-13 cells as they are a much more widely used and better-studied secondary AML cell line with frequent driver mutations. And most importantly, in contrast to MTA-restricted SKK-1 cells MOLM-13 are freely available from DSMZ. We explain our decision on page 7 (top paragraph). We have validated key findings from the SLAM-seq such as the down regulation of translation regulatory genes in SKK-1 cells (Figure 3H).

6. In Figure 5, the authors should provide a positive control for treatment by decitabine in the cells (e.g., demethylation of LINE-1 elements). They need to show the readers that their drug treatment results in the expected phenotypes.

Following the suggestion of this reviewer, we have confirmed that the used concentrations of decitabine are functional by showing a reduction of LINE-1 DNA methylation (Suppl. Fig. S4).

7. As the authors may know, we observe tremendous clinical utility in combining AZA with the BCL2 inhibitor venetoclax. What happens with venetoclax is introduced into their system? This is an important experiment. For example, could BCL2 have served as a positive control for their screen?

We thank the reviewer for this comment and do indeed agree that BCL2 would have been an excellent positive control in the screen. While we did not think of it at the moment of the screen design, which was primarily meant to represent a close to comprehensive chromatin-regulatory network, we have now included the Bcl2-inhibitor venetoclax in the series of experiments evaluating drug synergies (Figure 2C, last panel). Interestingly, in MOLM-13 cells the synergy observed on the level of cell survival between AZA and venetoclax was similar to that seen with CBP/p300 inhibitors.

8. The authors appear to only study two cell lines. Primary MDS cells should be used in validation experiments.

We fully agree that the major findings of this study should be validated in primary cells. To address this, we have performed co-treatment experiments in leukemic blast cells isolated from the bone marrow of AML patients (Figure 2E), in which we could confirm an increase in apoptotic cells in response to the combination of AZA and CBP/p300 inhibitors. Primary cell cultures are not part of our expertise and after failing to cultivate CD34+ cells from the bone marrow of MDS patients, we decided to switch to a limited number of bone marrow samples that were available from AML patients. In addition, we have expanded the panel of cell lines by including F36-P and MDS-L as two additional MDS-derived sAML cell lines (Figure 2D).

9. The authors should explain/contrast the cytogenetic/molecular mutations found in the SKK-1 and MOLM-13 cells used in the work. The authors cite their initial publication of the SKK-1 cell line, but it would be helpful for the reader if they briefly outlined the key features of these cell lines that make them relevant for these studies. The authors briefly describe these for the MOLM-13 cells, but in an off-handed way.

We apologize for not having better described the molecular characteristics of SKK-1 and MOLM-13 cells and their differences. A better description of both cell lines has now been included in the results section (page 5, top paragraph, and page 7, beginning of second paragraph). In the methods, we cite our previous study in which we provided a detailed comparative characterization of SKK-1, MOLM-13 and three other sAML cell lines using a panel of cytogenetic, cytometric and genetic methods (Palau et al., 2017, Genes Chromosomes Cancer).

10. A prior publication [Cheng, J.X. et al. RNA cytosine methylation and methyltransferases mediate chromatin organization and 5-azacytidine response and resistance in leukaemia. Nat Commun 21;9(1):1163, 2018. PMID: 29563491 PMID: PMC5862959] showed that RNA 5-methylcytosine (RNA:m5C) and the RNA:m5C methyltransferases (RCMTs) NSUN3 and DNMT2 directly bind hnRNPK, which interacts with lineage-determining transcription factors (TFs), GATA1 and SPI1/PU.1, and with CDK9/P-TEFb to recruit RNA-polymerase-II at nascent RNA, leading to formation of 5-Azacytidine (5-AZA)-sensitive chromatin structure. The group also showed that NSUN1 binds BRD4 and RNA-polymerase-II to form an active chromatin structure that is insensitive to 5-AZA, but hypersensitive to the BRD4 inhibitor JQ1. AZA-resistant leukemia cell lines and primary MDS/AML samples with AZA-resistance had increases in RNA:m5C and NSUN1-/BRD4-associated active chromatin. How can the authors reconcile their findings with this prior publication? Did their screen identify any of these molecules? [NSUN3 and BRD4 appear DNMT2 in Suppl Table 3, for example] Why is this publication- which seems directly relevant to their model- not cited/discussed?

We sincerely apologize for having omitted this relevant publication that has escaped our attention. We now present the study by Cheng et al. in the introduction when describing known differences between AZA and decitabine (page 4) and discuss it further together with our results in the discussion section (page 11).

To address the questions: In our screen, the shRNAs targeting NSUN3 were depleted in response to AZA treatment. This is in line with the in Cheng et al. described preferential inhibition of growth of AZA-sensitive leukaemia cells by downregulation of NSUN3. Furthermore, we saw an enrichment of BRD4 shRNAs after AZA treatment. This has now been added to the discussion. In the following figure that presenting the results of the shRNA screen these two genes have been highlighted.

Minor concerns:

1. "led" is misspelled in line 42.
2. In line 54, the authors should use the adverb "particularly", not the adjective "particular".

We apologize for these errors and have now corrected them.

3. In lines 57-58, the authors describe the use of hypomethylating drugs as being the treatment of choice for patients not eligible for allogeneic stem cell transplant. However, many patients are bridged to allogeneic stem cell transplant using these drugs, so this sentence does not really reflect how these drugs are used clinically.

We have modified the sentence to point out that these drugs have multiple clinical applications (page 3, paragraph 2).

REVIEWERS' COMMENTS

Reviewer #1 (Remarks to the Author):

The authors have addressed all my major points satisfactorily. After re-reading the manuscript, a small number of minor issues remains:

1. line 121: please provide a reference for AZA as a "chromatin-modifying agent" or use a more refined description.

2. line 156: please briefly explain the putative role of NAA15 in mediating AZA resistance in the discussion.

3. Fig. 2E: error bars need to be shown.

4. line 335-337: this statement needs to be properly referenced.

Reviewer #2 (Remarks to the Author):

Many of the comments from the initial review of the manuscript have been addressed.

The mechanistic model presented to address the issue of sensitivity to azacitidine being mediated through downregulation of CBP and subsequent effect on protein synthesis is superficial. Nothing in the model (figure 5E) directly links the use of azacitidine to CBP. For example, the model as currently depicted, can be interpreted to assume that azacitidine has an effect on protein synthesis that is independent of CBP, but that synergizes with CBP inhibition.

Authors need to address this issue to give the readership a clear picture of the contribution that this work brings to our understanding of how azacitidine works-i.e that sensitivity to azacitidine is mediated at least in part by downregulation of CBP/p300

Reviewer #1 (Remarks to the Author):

The authors have addressed all my major points satisfactorily. After re-reading the manuscript, a small number of minor issues remains:

1. line 121: please provide a reference for AZA as a "chromatin-modifying agent" or use a more refined description.

Thank you for this comment. We have referenced a review in which we explain the molecular actions of AZA in great detail (REF 11, Diesch et al. 2016). Furthermore, we hope that readers will find the detailed explanation of the drug's action useful that we provide in the introduction.

2. line 156: please briefly explain the putative role of NAA15 in mediating AZA resistance in the discussion.

We have now included a sentence suggesting the role of NAA15 in AZA resistance in the discussion.

3. Fig. 2E: error bars need to be shown.

Due to the limited cell numbers obtained from patient samples we were not able to perform these experiments using technical replicates, thus there are no error bars in Fig.2E. Furthermore, due to the heterogeneous nature of patient samples, we decided to show the results from three patients in three separate panels. We felt that this would be more appropriate than condensing them into a mean and its error.

4. line 335-337: this statement needs to be properly referenced.

We apologize for this omission. We have now referenced the first report quantifying the degree of AZA incorporations (REF 14, Li et al., 1970, Cancer Research).

Reviewer #2 (Remarks to the Author):

Many of the comments from the initial review of the manuscript have been addressed. The mechanistic model presented to address the issue of sensitivity to azacitidine being mediated through downregulation of CBP and subsequent effect on protein synthesis is superficial. Nothing in the model (figure 5E) directly links the use of azacitidine to CBP. For example, the model as currently depicted, can be interpreted to assume that azacitidine has an effect on protein synthesis that is independent of CBP, but that synergizes with CBP inhibition. Authors need to address this issue to give the readership a clear picture of the contribution that this work brings to our understanding of how azacitidine works-i.e that sensitivity to azacitidine is mediated at least in part by downregulation of CBP/p300

The mechanistic model has been included in response to a suggestion by Reviewer 1. The main aim of the mechanistic model was to highlight how CBP/p300 inhibition and AZA converge on protein synthesis. CBP inhibition leads to the downregulation of translation promoting genes. Independently of this, AZA is incorporated into RNAs and further reduces protein translation. The combination of both can reach synthetic lethality in cells. When presenting our data at conferences, we noted that the simplified model was helpful. Especially since in the clinical setting AZA is often referred to as hypomethylating drug, which is only one aspect of its molecular action. In conclusion, we suggest maintaining the model but to remove the blue line and question mark that are currently speculative.